# Myelodysplastic Syndrome: Diagnosis and Screening

**DOI:** 10.3390/diagnostics12071581

**Published:** 2022-06-29

**Authors:** Francisco P. Tria, Daphne C. Ang, Guang Fan

**Affiliations:** 1Section of Cellular Immunology and Molecular Pathology, Institute of Pathology, St. Luke’s Medical Center—Global City, Taguig 1634, Metro Manila, Philippines; franciscotria@yahoo.com (F.P.T.IV); daphnechuaang@yahoo.com (D.C.A.); 2Department of Hematopathology, Oregon Health & Science University, Portland, OR 97239, USA

**Keywords:** myelodysplastic syndromes, cytogenetics, next generation sequencing

## Abstract

Myelodysplastic syndromes (MDS) are heterogeneous groups of clonal myeloid disorders characterized by unexplained persistent peripheral blood (PB) cytopenia(s) of one or more of the hematopoietic lineages, or bone marrow (BM) morphologic dysplasia in hematopoietic cells, recurrent genetic abnormalities, and an increased risk of progression to acute myeloid leukemia (AML). In the past several years, diagnostic, prognostic, and therapeutic approaches have substantially improved with the development of Next Generation Sequencing (NGS) diagnostic testing and new medications. However, there is no single diagnostic parameter specific for MDS, and correlations with clinical information, and laboratory test findings are needed to reach the diagnosis.

## 1. Cytopenia

The secondary causes of PB cytopenia occur far more often than the primary BM neoplasms. Various nonclonal disorders affecting the BM, including viral (in particular retrovirus, parvovirus, hepatitis virus), bacterial and parasitic infections, autoimmune disorders (such as juvenile rheumatoid arthritis, polyarteritis nodosa, systemic lupus erythematosus, immune thrombocytopenic purpura), nutritional deficiencies (such as malnutrition, iron deficiency anemia, megaloblastic anemia due to vitamin B12 and folate deficiencies, vitamin D deficiency, zinc-induced copper deficiency, hypervitaminosis A), medication exposures, drugs, toxins (such as alcohol abuse, zinc, arsenic, chromium, cadmium), chronic kidney and liver diseases, and endocrinopathies, need to essentially be ruled out first, before a diagnosis of MDS can be made [1]. The diagnostic accuracy has practical consequences, and disagreements about the diagnosis are common [2].

Cytopenia is a “sine qua non” for any MDS diagnosis; however, specific cytopenias have only a minor impact on the classification. Moreover, the lineage manifesting significant morphologic dysplasia frequently does not correlate with the specific cytopenia in individual MDS cases [3].

The recommended threshold of cytopenia as per the risk stratification guidelines, defined in the International Prognostic Scoring System (IPSS), include hemoglobin of <10 g/dL, a platelet count of <100 × 10^9^/L, and an absolute neutrophil count of <1.8 × 10^9^/L [4,5]. However, MDS may still be diagnosed with a milder degree of cytopenia (hemoglobin <13 g/dL in men or <12 g/dL in women, or a platelet count of <150 × 10^9^/L), if unequivocal morphologic dysplasia and/or cytogenetic abnormalities are present [6]. In addition to these, several other factors, such as ethnicity, age, sex-related differences, and altitude, are also valuable factors and need to be considered [7].

## 2. Morphology

The morphologic classification of MDS practically depends on several diagnostic approaches, which include the presence of an increased blast population in both the BM and PB, the evaluation of cytologic abnormalities and dysplastic changes among the hematopoietic elements, the evaluation of cellular topography and cellularity, and the presence or absence of fibrosis.

Traditionally, the World Health Organization (WHO) defined the MDS categories depending on the blast percentage in the PB and BM, but a threshold of less than 20% is a mandatory cut-off. The presence of an increased blast population of 2–4% in PB or 5–9% in BM is categorized as MDS with excess blast 1 (MDS-EB-1), whereas the presence of a higher number of blasts, with 5–19% in PB or 10–19% in the BM, or the presence of unequivocal Auer rods by morphologic examination are categorized as MDS with excess blast 2 (MDS-EB-2) [6]. However, the WHO stated that the 20% blast cut-off is not a mandate to treat the patient as having AML or blast transformation, and that the therapeutic decisions must always be based on the clinical situation, after all of the information is considered [8]. For this reason, a blast count of 10–30% is suggested for enrolment in either MDS or AML clinical trials.

Morphologic dysplasia may occur in one or more of the hematopoietic lineages. Erythroid dysplastic features occur in the form of megaloblastoid changes, multinuclearity in erythroid precursors, nuclear lobulation, the presence of pyknosis or chromatin condensation in the nuclei, cytoplasmic fraying involving 50% of the cellular membrane, internuclear bridging, cytoplasmic vacuolization, and the presence of ringed sideroblasts described as five or more granules encircling one third or more of the nucleus by iron stain (Figure 1A). On the other hand, dysplastic features in the granulocytic lineage can occur in the presence of myeloblasts and Auer rods, pseudo-Pelger–Hüet changes or nuclear hyposegmentation and pseudo-Chediak–Higashi cytoplasmic inclusions, abnormal nuclear irregularities, and abnormal cytoplasmic granules and cytoplasmic hypogranulation (Figure 1B). The megakaryocytic dysplastic features are described as micromegakaryocytes, small mononucleated, binucleated megakaryocytes, and the presence of separated nuclei (Figure 1C) [2,9]. It should be noted that the spectrum of dysplastic changes may range from mild changes to overtly bizarre abnormalities. The WHO stated that at least 10% of the cells in a given lineage should be dysplastic to qualify as a significant finding; however, the interobserver variability is more problematic in cases where the degree of dysplasia is near to the 10% threshold. This cellular enumeration should be performed in a 200 PB leukocyte count and a 500 BM nucleated cell count [3]. The 10% cutoff was meant practically to prevent the inclusion of mildly odd-looking hematopoietic cells observed in healthy, elderly populations. It should also be noted that morphologic dysplasia is a common occurrence in the bone marrow of healthy individuals [10]. That is why clinical history, cytogenetics, and molecular studies are needed for a MDS diagnosis.

## 3. Flow Cytometry

Alterations in the hematopoietic stem cells were generally regarded as one of the disease-initiating events in MDS, either in primary de novo disease or secondary therapy-related causes, both of which ultimately result in disrupted hematopoiesis. Even though the morphology is considered to be one of the key features in the diagnosis, flow cytometry (FC) has also been considered to be an important tool for the diagnosis, prognostication, and monitoring of the disease course [11]. The International and European LeukemiaNet Working Group (ELN IMDS Flow WG) Guidelines suggested the standardization of protocols for using FC in MDS. A bulk lysis method of mature erythrocytes should be performed before the cells are incubated in a panel of antibodies, and the acquisition of a minimum of 100,000 CD45^+^ events is optimal [12,13]. The cellular dysplastic features can reflect the surface antigen aberrancies, which can be detected using multiparametric FC (MFC). The utility of MFC in MDS focuses on the assessment of the precursor myeloid antigen aberrancies, abnormalities in the myeloid maturational patterns among the granulocytic and monocytic lineages (Figure 2A,B), the enumeration of blast immunophenotypes, and, to some extent, the decrease in the progenitor B-cells (hematogones) [14]. There is no specific FC marker for documenting MDS, but multiple immunophenotypic abnormalities can establish the clonality of the disease. The presence of multiple antigenic aberrancies has been shown to have a higher predictive value for MDS as compared to a single aberrancy, and, therefore, findings of multiple antigen expression anomalies might strongly support the diagnosis of MDS by FC [15,16]. The analysis of the antigenic parameters can be achieved using a minimum requirement of a four-color FC, and analysis should be focused on the immature myeloid progenitor cells. Looking into the following parameters allows for the differentiation of normal and abnormal progenitor cells: (1) measurement of progenitor cells; (2) evaluation of their progenitor cell plasticity (CD34 and CD45 expressions), in combination with forward and side light scatter (FSC and SSC) plots; (3) expression of CD117; and (4) expression of maturation and lineage infidelity markers [13].

A four-parametric scoring system (referred to as the Ogata score) is widely used as a simple FC criterion for MDS diagnosis, which contributes only a minimal inter-observer variability, and with a reported specificity of 93% and sensitivity of 70% [17,18]. The parameters described in this scoring scheme are as follows: (1) SSC of neutrophils defined as the ratio of lymphocyte SSC (granulocyte/lymphocyte SSC); (2) percentage of CD34^+^ myeloid precursors among the entire viable nucleated cells (% of CD34^+^ myeloblasts); (3) percentage of CD34^+^ precursor B-cells (hematogones) among the entire CD34^+^ cells (% of CD34^+^ B-cells); and (4) CD45 antigenic expression of CD34^+^ myeloid progenitors as ratio to lymphocytes (lymphocyte/myeloblast CD45 ratio). The established reference values were identified, and deviations from the references are scored as one point with a maximum of four points. A score of ≥two points indicated MDS, and high scores of three or four are associated with a high probability of MDS, [17,18]. However, this scoring system is less applicable in the pediatric cohort presenting with refractory cytopenia [19]. 

Since dyserythropoiesis is a common feature of MDS, the integration of erythroid aberrancies in flow cytometric panels led to an increased sensitivity in detecting MDS. Aside from evaluating antigen expression aberrancies on immature myelo/monocytic cells in MDS, the analysis of immunophenotypic aberrancies of nucleated erythroids further aid in supporting the diagnosis of MDS [20]. The ELN IMDS Flow WG has demonstrated the utility of antibody markers, comprising of CD45, CD36, CD71, CD105, CD117, and CD235a, which can be used to evaluate the erythroid dysplasia, and produce an erythroid score (RED score) which evaluates the aberrant erythroid antigen expression and hemoglobin levels [20]. The IMDS flow group proposed guidelines for erythroid evaluation which are: (1) CD36 coefficient of variation (CV); (2) CD71 CV; (3) mean fluorescence intensity (MFI); and (4) CD117 positive within CD45 negative-diminished cell fraction (percentage of progenitors). The increased CV of CD71 has been the most sensitive marker for MDS, followed by an increased/decreased percentage of CD117, while an increased CV of CD36 has been reported to be the most specific marker [21]. The analysis of the antigen aberrancies in the erythroid markers CD71 CV and CD36 CV, and an abnormal percentage of CD117^+^ erythroid progenitor cells, have been reported to provide the best discriminating parameters between MDS and non-clonal cytopenias [20]. It has also been reported that presence of multiple erythroids aberrancies are significantly correlated with MDS [21].

Additionally, many of the investigations of the immunophenotypic alterations identified by MFC can lead to discoveries of newer targets for future drug development [11].

## 4. Cytogenetics

Cytogenetics has been an important and a necessary parameter in the diagnosis of MDS. The WHO relies heavily on cytogenetic aberrations in MDS. In addition to establishing a clonal process in patients with peripheral blood cytopenia, cytogenetics plays a major role in the prognostication, clinical-morphologic correlation, theragnostic strategies, and in predicting the likelihood of progression to AML. In contrast to other myeloid malignancies, in which the diagnosis is defined by a single cytogenetic event (such as chronic myeloid leukemia and acute promyelocytic leukemia), there is a vast spectrum of cytogenetic-defining lesions in MDS, making diagnosis very challenging. Nevertheless, roughly 50% of MDS has normal cytogenetics. The cases with borderline dysplasia and normal cytogenetics present with diagnostic challenges. Various combinations of chromosomal lesions contribute to the large spectrum of clinicopathologic features of MDS. According to the WHO, MDS may still be diagnosed in a patient with unexplained cytopenia, when all of the other secondary causes of refractory cytopenia are essentially and exhaustibly ruled out, and as long as there are MDS-defining cytogenetic abnormalities identified (with some exceptions) [6].

The results from cytogenetic studies, both by conventional karyotyping (G-banding) and fluorescence in situ hybridization (FISH) assays, serve as strong parameters included in the revised International Prognostic Scoring System (IPSS-R) score for MDS. As such, the IPSS-R has been proven to be beneficial for predicting the clinical outcomes for untreated MDS patients and aiding in the design of clinical trials for the disease [5]. The comprehensive cytogenetic scoring system (CCSS) was adapted by the WHO, which defines the specific prognostic stratification of MDS based on the existing cytogenetics clone (Table 1). The IPSS-R encompass five cytogenetic subgroups, which gives more weight to chromosomal aberrations than the previous IPSS (Table 2). The five cytogenetic risk groups were defined according to the new CCSS, that was based on a large multicenter database. Overall survival (OS) was reported to be significantly different independently from the cytogenetics’ status [22].

In contrast to AML, balanced structural abnormalities, such as translocations and inversions, are relatively rare in MDS [23]. In general, MDS show a characteristic, overwhelmingly higher prevalence of unbalanced chromosomal abnormalities. More often, these genetic lesions tend to occur in the form of the chromosomal loss of genetic material, such as deletions and monosomies, with a lesser frequency of a gain of genetic material in the form of trisomies (Table 3). A huge number of the cases also occur among those with complex cytogenetic alterations (with three or more abnormalities).

Sequential cytogenetic analysis (both karyotyping and FISH) of follow-up samples has been established as identifying the possible clonal evolution of MDS clones leading to greater understanding of the heterogenous acquisition or loss of these genetic events in MDS [24]. Demonstrating an occurrence of clonal evolution and, to a lesser degree, the presence of subclones, at the time of diagnosis in patients with MDS, has an unfavorable influence on survival and AML progression. An altered karyotype at the baseline appears to predispose toward the acquisition of further cytogenetic alterations. Nevertheless, no specific patterns of clonal evolution emerged according to the baseline karyotypic anomalies [25].

### 4.1. del(7q) or Monosomy 7

Monosomy 7 (-7) and deletions of the long arm of chromosome 7 (del(7q)) are found in several myeloid neoplasms, suggesting its crucial role in disease pathogenicity. They occur either in isolation or as part of a complex karyotype, and are generally associated with unfavorable prognosis in certain disease entities. In MDS, the isolated cytogenetic abnormality of del(7q) has been categorized in the intermediate prognostic subgroup, whereas isolated cytogenetic abnormality of -7 has been classified as belonging in the poor prognostic subgroup. Among the patients with del(7q), there was a tendency toward better survival compared with the patients with complete -7 as an isolated abnormality and as a noncomplex aberration. However, because the survival difference between these related cytogenetic lesions are not statistically significant, they are then regarded as a single cytogenetic category [26]. The chromosome 7 anomalies are reported in approximately 10% cases of de novo MDS, and up to 50% of therapy-related MDS [27].

The deletion breakpoints in chromosome 7 are heterogeneous and the deletions are often interstitial. The majority of the cases had proximal breakpoints in 7(q11) or 7(q22). The commonly deleted regions on 7q identified in MDS are located at positions 7q22, 7q32-33, and 7q35-36 [28]. Monosomy 7, occurring as the sole cytogenetic anomaly in a small but significant number of cases, may denote a dominant mechanism involving critical tumor suppressor gene(s) [29]. Previous studies identified possible driver genes contributing to the pathogenesis of -7/del(7q), including *CUX1, EZH2, LUC7L2, MLL3,* and *SAMD9/9L* [30]. However, specific therapies have not yet been developed.

*CUX1* is a conserved, haplo-insufficient tumor suppressor frequently deleted in myeloid neoplasms. It encodes a homeodomain-containing transcription factor, which is located in chromosome band 7q22.1. In the RNA-sequencing data, a *CUX1*-associated cell cycle transcriptional gene signature was identified, suggesting that *CUX1* exerts tumor suppressor activity by regulating the proliferative genes [31].

*EZH2* acts as a tumor suppressor for myeloid malignancies. It is located at chromosome 7q36, which encodes a member of the polycomb group family, that forms multimeric protein complexes which are involved in maintaining the transcriptional repressive state of genes. It encodes a histone methyltransferase which functions in the epigenetic silencing of the genes involved in stem cell renewal [32]. However, deletions in 7q do not result in the loss of the *EZH2* gene [33].

### 4.2. del(5q)

The 5q- syndrome was first reported in 1974 [34]. Currently, the only subtype of MDS defined by a genetic abnormality is the group with an isolated deletion in the long arm of chromosome 5 (del(5q)). This specific type of MDS belongs to the good prognostic subgroup. The most common symptom is usually of macrocytic anemia, with thrombocytosis a more common occurrence than thrombocytopenia [6].

The most common abnormality includes the interstitial deletion of the long arm of chromosome 5. Larger losses of the 5q arm, by deletion of the centromeric or telomeric regions or mutations involving the *NPM1* or *MAML1* and *APC* genes, have been related to a higher risk of MDS and an earlier risk of transformation to AML [35,36]. The MDS with isolated del(5q) are the most common genetic abnormality seen in de novo MDS and they have a relatively better prognosis and a reduced risk of transformation to AML [4]. However, this abnormality may be a part of complex cytogenetics, in which the prognosis of these cases is poorer. From these MDS cases, del(5q) is not necessarily a primary genetic event, and this may be acquired after other disease-initiating mutations, particularly the epigenetic modifier mutations [37].

It was discovered that the haploinsufficiency of several genes located in this region are capable of generating the clinical phenotype in patients with MDS. The loss of one *RPS14* allele, for example, can recapitulate the dyserythropoiesis seen in MDS del(5q). The loss of this protein has been shown to upregulate *p53*, primarily in erythroblasts, and therefore to promote apoptosis from these cells. The mutations of *p53* are significantly associated with the loss of del(5q) and a complex karyotype, and this has not been associated with del(7q) [38]. Mutations in *RPS14* are also found in about 25% of the patients with Diamond–Blackfan Syndrome, which leads to haploinsufficiency of *RPS14* [39]. The haploinsufficiency of several other genes of the commonly deleted region include *HSP9*, *CTNNA1*, and *EGR1* [35]. On the other hand, the loss of one copy of the microRNA miR-145 and miR-146 leads to the presence of preserved or even increased levels of the platelet count observed in MDS patients, with isolated del(5q) [40]. The loss of these microRNAs leads to the upregulation of *TRAF6,* resulting in thrombocytosis, neutropenia, and megakaryocytic dysplasia [41].

Lenalidomide is a thalidomide analog, which shows a dramatic therapeutic effect in patients with low-risk MDS. Its response rate was significantly higher among those with interstitial deletion involving chromosome 5q [39]. It can also reduce the transfusion requirement and can reverse cytologic and cytogenetic abnormalities in MDS with del(5q) [42].

### 4.3. del(20q) and Loss of Y

The deletion of the long arm of chromosome 20 (del(20q)) is a recurrently identified cytogenetic abnormality in myeloid neoplasms, including myeloproliferative neoplasms (MPN), MDS/MPN, MDS, and AML. However, unlike del(5q), del(20q) is not recognized by the WHO as a unique entity in MDS. Del(20q), as an isolated cytogenetic abnormality, can be seen in the bone marrow specimens of patients without morphologic diagnostic features of any of the myeloid neoplasms, and this may also be seen in patients with non-myeloid malignancies or unexplained cytopenias. Therefore, the WHO emphasizes that the presence of isolated del(20q) is not considered to be definitive clonal evidence of MDS in patients with unexplained cytopenia, in the absence of morphologic evidence of MDS [6]. This may impose a diagnostic dilemma on several MDS cases. The diagnostic samples of patients with isolated del(20q) without mutations have a very low risk for progression to a myeloid neoplasm, and approximately one third of the patients with mutations ultimately progressed into a myeloid neoplasm [43].

MDS with isolated del(20q) are categorized in the good cytogenetic prognostic subgroup by the IPSS-R. The breakpoints of del(20q) are heterogeneous. The most frequently mutated genes identified include *U2AF1*, *ASXL1*, *SF3B1*, *TP53*, and *SRSF2*. In MDS, del(20q) may cause deletion of the *ASXL1* gene, and *ASXL1* alteration exerts a negative impact on MDS with del(5q). This has also been correlated with a lower platelet count and a poor response to azacytidine (AZA) [44]. The *ASXL1* mutations are found in 11 to 21% of patients with MDS and are a predictor of poor OS [45,46,47]. The median survival of patients with del(20q) was 54 months, compared with 12 months in the patients with del(20q) plus other additional chromosomal abnormalities [48].

The deletion of the Y chromosome (-Y) belongs to the very good prognostic subgroup in MDS, as categorized by the IPSS-R, but has also been attributed to an age-related phenomenon. It is noteworthy that, although it has been associated with better prognosis in MDS, the exact mechanism remains unknown. It was also known that the loss of X in women and the loss of Y in men increases with age [49]. Deletion of the Y chromosome has been observed in about 4 to 10% of male patients as a single cytogenetic abnormality in MDS [22]. Although -Y influences the prognosis in MDS, it occurs in elderly men with no evidence of hematologic disease. It was evident that the CD34+ cells carrying -Y are more prevalent in male patients with MDS than the healthy counterparts [50].

### 4.4. Trisomy 8

Trisomy 8 (+8) is the most common chromosomal gain in MDS, and is seen in about 11% of de novo MDS with an abnormal karyotype [51]. It belongs to the intermediate prognostic subgroup categorized by the IPSS-R. Although it is a common cytogenetic abnormality, the presence of isolated +8 is not considered as presumptive evidence of MDS without the minimal dysplastic morphologic criteria. One of the main reasons is that +8 can be found as a constitutional mosaicism in healthy individuals [52]. Several reports indicate that the presence of +8 is constitutional in 15 to 20% of MDS and acute leukemia patients [53]. Another reason that the sole presence of +8 is not a MDS-defining event is that it is also seen as a clonal aberration in aplastic anemia, which may also be a close differential diagnosis of MDS, but disappears after immunosuppressive therapy [54]. Therefore, the presence of unequivocal morphologic dysplasia is required to discriminate hypoplastic MDS from aplastic anemia. It was also noted that MDS with +8 responds well to immunosuppressive therapy, with an up to 67% response rate [55].

### 4.5. del(11q) and del(12p)

MDS with isolated del(11q) is associated with a very good prognosis (similar to -Y) characterized by the IPSS-R. It is a rare cytogenetic abnormality, which is reported to occur in 0.7% in de novo and secondary AML and MDS [56]. The *KMT2A* gene (previously referred to as *MLL* or mixed lineage leukemia gene) is located at the 11q23 breakpoint. It was shown that del(11q) is heterogenous at the molecular level and may signify cryptic rearrangements involving chromosome 11 or the *KMT2A* gene. Unlike in AML with del(11q), which harbors the cryptic *KMT2A* rearrangements, it was reported that MDS with del(7q) lacks this cryptic rearrangement, and therefore may potentially explain the biological differences between AML and MDS with del(11q) [57].

Another gene located at chromosome 11q23 telomeric to the *KMT2A* gene is the *CBL* gene. Being a signal transduction gene, the mutations in the *CBL* gene constitute important pathogenic lesions associated with AML progression, due to the impaired degradation of the activated tyrosine kinase [58,59].

Deletion in the short arm of chromosome 12 is a rare event in MDS, and occurs in 0.6 to 5% of the cases at diagnosis [60]. This was categorized in the good prognostic subgroup in the IPSS-R with an OS of 76 months [5]. It usually occurs as a very small interstitial deletion between 12p12.2 and 12p13.1, affecting the *ETV6/TEL* gene [61]. However, the *ETV6* deletion is seen to be higher in AML than in MDS [62].

### 4.6. del(9q)

The deletions in the long arm of chromosome 9 are more commonly seen in AML than in MDS or MPN. The IPSS-R categorizes del(9q) in the intermediate risk subgroup with a median OS of 32 months. The myeloid neoplasms with del(9q) were identified as having a high prevalence of *TET2* mutations, and the association was more pronounced when *TET2* was the sole abnormality, with a frequency of 45% [63]. Recent data showed that del(9q) was removed from the list of MDS-defining cytogenetic abnormalities, because of its association with t(8;21) and the frequent occurrence in AML with *NPM1* and biallelic *CEBPA* mutations [64].

### 4.7. t(17p) or Isochromosome 17q

The presence of a chromosome 17 abnormality in MDS has been correlated with poor prognostic features and very low OS, except for isochromosome 17q (i(17q)), which is associated with the intermediate risk prognostic subgroup by the IPSS-R. The association between a poor prognosis and a chromosome 17 abnormality in patients has also been found in the context of a complex karyotype. This was also correlated with the loss of 17.13.1, which contains the genetic loci of the tumor suppressor gene p53 (*TP53*) [65]. The significance of this cytogenetic aberration has been valuable in MDS and AML with *TP53* mutations, due to its favorable response to hypomethylating agents (HMA), particularly decitabine [66]. *TP53* is the most commonly mutated gene in human cancer. It functions as a transcription factor for cell cycle arrest, DNA repair mechanisms, apoptosis induction, and cellular differentiation. In MDS, the *TP53* mutation was significantly associated with del(5q) syndrome, with its diverse roles in cell cycle, DNA repair and apoptosis leading to chromosomal instability, and AML transformation [67]. The association of *TP53* in the pathogenesis of MDS was also seen in the context of therapy-related MDS (t-MDS), as defined by the WHO’s classification in 2016, wherein an exposure to cytotoxic or radiation therapy for a previous unrelated malignancy or autoimmune disease was documented [6].

### 4.8. t(11;16)

The balanced translocation of chromosome 11 and 16 occurs in approximately 3% of the therapy-related MDS cases [6]. The *KMT2A* gene (previously known as the *MLL* gene) has been mapped in the 11q23 locus and this gene forms fusion transcripts with more than 70 translocation partner genes. The *KMT2A* gene translocation results in the formation of a chimeric protein in its amino-terminal and fused in the carboxy-terminal portion of the fusion partner gene [68]. On the other hand, the *CBP* gene encodes a transcriptional adaptor/coactivator protein in the 16p13 locus, and is involved in the regulation of the cell cycle [69]. It was postulated that one possible explanation for the leukemogenesis of t(11;16)-positive MDS is the loss of function in *CBP* to regulate the cell cycle by its structural alteration when fused with *KMT2A* [70]. The OS of adult patients with t(11;16) in one study was similar to adult patients with therapy-related myeloid neoplasms and complex karyotypes [70].

### 4.9. inv(3) or t(3;3)

MDS with inversion 3 (inv(3)) and balance translocation of chromosome 3 (t(3;3)) are observed in approximately 1% of the cases, and has been categorized in the poor prognostic subgroup [6]. Myeloid neoplasms with inv(3)/t(3;3) often present with anemia, and platelet counts which may be normal or increased [71,72]. The chromosomal aberration involves protooncogene *EVI1* at 3q26.2.2 or the longer form *MECOM* and *RPN1*, resulting in ectopic and overexpression of *EVI1*, or *MECOM* or the *RPN1/EVI1* fusion transcripts, with *RPN1* acting as an enhancer of *EVI1* expression [73,74]. *EVI1* has been associated with several signaling pathways, leading to cell growth, cell differentiation impairment, and cell survival [75]. *GATA2* was also implicated and was observed to be overexpressed in these cases, suggesting its role in the development of chromosome 3 rearrangements [74].

### 4.10. t(6;9)

The translocation of chromosome 6 and 9 (t(6;9)) is a rare occurrence in MDS, occurring in 1% of all of the MDS cases [6]. The translocation results in the formation of a chimeric fusion protein *DEK/NUP214* in der(6). This cytogenetic event has been associated with a poor prognosis in myeloid neoplasms. This abnormality is predominantly occurring as a sole karyotypic aberration, but a subset has been associated with a complex karyotype [76]. In AML, there is a high occurrence of *FLT3-ITD* mutations in patients with t(6;9) [76]. It was suggested that MDS with t(6;9) does share some clinicopathologic features with AML with t(6;9), which include comparably low hemoglobin levels, the presence of multilineage dysplasia, and some mutational landscape, however, it was also suggested that MDS cases are prognostically not equivalent to AML [77].

## 5. Next Generation Sequencing (NGS)

Over the past decade, the explosion of identified molecular signatures in MDS has been defined, with the development of high throughput sequencing studies. These major technological advances and breakthroughs in genetics unraveled the previously hidden mysteries of disease drivers that led to a drastic paradigm shift, not only in disease classification, but also in prognostication and therapeutic guidelines in myeloid malignancies. In the most recent WHO classification, clonal evidence in MDS mainly focuses on cytogenetic alterations; however, approximately 40 to 50% of MDS cases have normal cytogenetics, leaving diagnostic challenges for the diagnosing physician. This has been complicated by the discovery of clonal hematopoiesis of indeterminate potential (CHIP), wherein certain somatic pathogenic leukemia-associated variants are identified in healthy elderly individuals who have never had any evidence of cytopenia or cytologic dysplasia in the bone marrow or peripheral blood samples [78,79,80]. However, the presence of CHIP mutations increases the risk of progressing not only to primary hematologic neoplasm (Figure 3A,B), but also in other non-hematopoietic systemic diseases. The risk of progression of developing a hematologic malignancy in the presence of CHIP is only 0.5% to 1% per year, and therefore the vast majority of individuals with CHIP never develop an overt hematologic disease [78].

Aside from CHIP, other entities have been described as being potential pre-MDS conditions. A spectrum of clinico-pathological and molecular entities has been continuously reported in the literature and have been adapted in daily clinical practice. Idiopathic Cytopenia of Undetermined Significance (ICUS) is defined as the presence of persistent and unexplained peripheral cytopenia(s) in one or more lineage with no documented evidence of MDS-related mutations, while failing to demonstrate significant cellular dysplasia and marrow blast count of <5%. ICUS was also categorized into four subgroups, based on which particular cytopenic lineage is affected. Entities such as ICUS-A (anemia), ICUS-N (neutropenia), ICUS-T (thrombocytopenia), and ICUS-PAN (bi/pancytopenia) have been used. It was reported that ICUS-PAN may harbor a higher risk of transformation to an overt hematologic neoplasm, as compared to ICUS-A or ICUS-N [81]. The diagnosis of ICUS should only be made when all of the potential differential diagnoses have been exhaustively excluded [82]. As soon as one or more MDS-associated somatic mutations are detected in either PB or BM, the diagnosis of Clonal Cytopenia of Undetermined Significance (CCUS) should be made, provided that the diagnosis of an overt MDS falls short. This means that CCUS is defined as the presence of refractory cytopenia, in association with a detectable MDS-related mutation but significant morphologic dysplasia is not apparent. It is also very important to recognize that patients with CCUS showed a significantly higher probability to develop an overt myeloid neoplasm, compared to ICUS [83]. Another condition described within this spectrum, and which was adapted in clinical practice, is Idiopathic Dysplasia of Uncertain Significance (IDUS). This condition is defined as the presence of significant morphologic dysplasia of about ≥10%, in either myeloid, erythroid, and/or megakaryocytes, without a reported history of peripheral cytopenia, and therefore also falling short of a MDS diagnosis. A diagnosis of IDUS should only be made once reactive secondary and systemic chronic conditions and non-clonal entities, such as vitamin/nutritional deficiencies, drug toxicity, and infection, among others, have been ruled out [81].

It has been demonstrated that approximately 10% of what are supposed to be healthy individuals aged 70–80 years carry one or more somatic variants indicating the presence of CHIP. The specific variants, particularly those in the *DNMT3A*, *TET2* and *ASXL1* genes (*DTA* genes), have largely been implicated with CHIP, especially with a variant allele frequency (VAF) of ≥2% [78,79]. In addition to the *DTA* mutations, other genes have also been implicated with CHIP mutations, such as *RUNX1*, *IDH1/2*, and *JAK2,* among others. On the contrary, certain spectrum of the mutations in AML have already been implicated to possibly arise from an underlying MDS clone, or after a leukemogenic therapy (t-AML), in particular, the presence of the splicing gene mutations (*SRSF2, SF3B1, U2AF1, and ZRSR2*), has been established to be more than 95% specific for the diagnosis of secondary AML (s-AML) [84].

The growing interest in and understanding of the complex biology and genetic sequences of genes have substantially changed the diagnosis and treatment of MDS. Although there is no molecular signature that is entirely specific and sensitive for the diagnosis of MDS, and although several mutations seen in CHIP are also diagnostic for MDS, many of these gene mutations have significant predictive and prognostic values in MDS. The *DNMT3A* mutation is present in approximately 15% of MDS cases, and its presence is associated with a shorter OS in de novo MDS, as well as a higher tendency to evolve to sAML [37,46,86]. *TET2* is another recurrently mutated gene, identified especially during the early events of MDS pathobiology, and is reported to occur in about 20–30% of the cases [37,87]. The mutations in the *TET2* gene also provide significant therapeutic value. The response rate using HMAs, such as AZA and decitabine (DEC), were reported to be higher in the subset of MDS patients with *TET2* mutations compared to the *TET2* wildtype, especially when the major disease clone is identified [86]. However, despite its therapeutic implications, OS was not significantly correlated [88]. The *ASXL1* mutation is another epigenetic modifier which is identified in approximately 10–20% of MDS, with frameshift mutations and heterozygous point mutations being identified in about 70% and 30%, respectively, both of which are associated with adverse prognostic outcome and shorter OS [37,88,89]. *IDH1* and *IDH2* are seen in 5–12% of MDS patients and are thought to also represent an early driver of disease progression [89,90,91]. The relatively low frequency of the occurrence of *IDH* variants suggests that it is less frequently involved in the ancestral myeloid neoplastic clone [92]. It was found out that the *IDH1*/2 mutations in MDS are associated with lower absolute neutrophil count, higher bone marrow blast percentage, higher platelet counts, and with survival not significantly different when comparing *IDH1* vs. *IDH2* [90]. However, the presence of *IDH* mutations is associated with an unfavorable prognosis in MDS [93]. Emerging data have shown the efficacy of novel agents, including targeted *IDH1* (ivosidenib) and *IDH2* (enasidenib) inhibitors for cytoreduction for patients with high-risk MDS, who have been refractory with HMA inhibitors [94].

Aside from epigenetic modifying genes described above, splicing gene mutations are reported to occur in almost half of MDS cases and are strikingly common in those with normal cytogenetics (CN-MDS), and often represent early events in the pathogenesis [37,58,95]. *SF3B1* is the most frequently mutated gene reported in MDS, occurring in about 25–35% of patients [37,87]. It has been correlated with particular characteristic cellular morphology and associated with a significant favorable prognosis and high response rate to luspatercept [96].

A spectrum of other molecular pathways has also been reported and have significant biological affects in clinical behavior and disease progression in MDS. Transcription factors (such as *BCOR*, *KMT2a*, *RUNX1*, *WT1,* and *CREBBP*), signaling gene mutations (such as *KRAS*, *NRAS*, and *CBL*), and repair mechanism pathways (such as *TP53*) have all been described [37,58,88].

At least one or two somatic mutations are identified in MDS (Figure 4) [37,58,88]. Currently, somatic mutations in the *SF3B1* gene have been included by the WHO in stratifying a subset of MDS cases, which has been categorized to have an excellent prognosis and has been implicated with the presence of ring sideroblasts. The cases of MDS with ring sideroblasts (MDS-RS) traditionally require 15% ring siderblasts from the entire erythroid precursors; however, in the presence of the *SF3B1* mutation, the cutoff requirement goes down to only 5% [6]. It was found out that the specificity of the *SF3B1* mutation in MDS is 0.97, either as an isolated mutation or in the presence of other co-mutations [83].

The outcomes of MDS patients progressively worsen as the number of the acquired somatic mutations increases. A number of these somatic mutations have been implicated in predicting a lower OS in univariate analyses, in particular *TP53*, *EZH2*, *ETV6*, *RUNX1*, *ASXL1*, and *SRSF2*, whereas the presence of *SF3B1* has been implicated in better outcomes [46,87]. Aside from *SF3B1* (associated with a better outcome) and *TP53* (associated with a worse outcome), there is no other defined variant that has been implicated in having an independent prognostic value. The presence of mutations in either *SRSF2* and *U2AF1,* or the co-mutation of both, showed 100% specificity for myeloid neoplasm in the presence of myelodysplasia [83].

However, currently other mutations (aside from *SF3B1*) have not yet been concretely included by the WHO as evidence of clonality or for the prediction of outcome in MDS.

Mandatory morphologic and cytogenetic work-up is required in all of the cases with cytopenia of an unknown secondary etiology. Nevertheless, the identification of driver mutational signatures by NGS in these cohorts of patients certainly helps to determine clonality, especially in those with equivocal morphologic dysplasia and those with normal cytogenetics [58]. Besides the specific recurrent gene mutations in MDS, the variants with higher VAF are more supportive of a MDS diagnosis than CHIP [85]. For borderline cases of MDS and AML, with blast counts nearing the 20% cutoff, NGS can further support the AML diagnosis if certain molecular findings, such as mutations in the *NPM1*, *FLT3* and *CEBPA* genes, are detected. The NGS findings can also aid in diagnosing cases of hypocellular marrow with peripheral blood cytopenia and in discriminating between aplastic anemia and hypoplastic MDS.

Recurrent cytogenetic aberrations have been implicated to be MDS-defining in patients with cytopenia, with or without an overt morphologic dysplasia (except for sole abnormality involving the gain of chromosome 8, del(20q), and the loss of the Y chromosome) (Table 3). In a subset of patients with prolonged and refractory cytopenia and normal cytogenetics (CN-MDS), and a high clinical suspicion of MDS, 91% of CN-MDS demonstrate an underlying mutational signature, which clearly can provide clonal evidence of a myeloid malignancy and confirm a MDS diagnosis [58]. Recent data show that NGS can not only aid in the demonstration of a clonal event in patients with cytopenia in MDS, but can also predict and help prognosticate the CN-MDS cases. The data show that 64% of CN-MDS patients with histone modifier mutations and signal transduction gene mutations have been associated with AML transformation at relatively earlier timepoints, as opposed to the histone modifier and signaling genes wildtype (Figure 5A,B), and therefore clearly shows that the mutational landscape which can be demonstrated by NGS should be included in the WHO definition of MDS [58].

The traditional understanding that the presence of higher VAF levels at diagnosis has been implicated to be the early acquired somatic clones in MDS or AML, and therefore CHIP mutations have been proven to be one of the earliest events in the pathogenesis of these myeloid neoplasms. This process can lead to further genetic instability, and make the neoplasms more prone to acquisition of additional mutations until a malignant driver clone is acquired (Figure 3A). The emerged dominant mutant clones that are acquired late in the pathogenesis are oftentimes implicated in the AML transformation. During the course of the disease spectrum, the clonal evolution can also be tracked and observed through sequential NGS follow ups among these cases. 

Whole exome sequencing studies reveal complex and novel pathways in MDS. The frequent mutations of the spliceosome machinery were also identified to be one of the most frequently affected pathways in MDS, particularly the mutations in *SF3B1*, *SRSF2*, *U2AF1*, and *ZRSR2* [97].

Aside from the novel DNA mutations, reciprocal balanced gene rearrangements can now be detected by the NGS platforms. The AML-defining recurrent genetic alterations, involving *PML-RARA*, *RUNX1-RUNX1T1*, and *CBFB-MYH11,* can now be detected through targeted RNA sequencing, which may further aid in discriminating between AML and MDS in equivocal blast-count cutoff threshold.

With a multitude of detectable mutational markers in MDS and AML, NGS offers the opportunity and leverage for measurable residual disease (MRD) monitoring, especially in the presence of somatic driver mutations (such as *NPM1*, *CEBPA*, *RUNX1*, *SF3B1*, and others) at diagnosis. NGS generally can detect as low as 1% VAF, and therefore can play a complementary role with targeted quantitative real-time PCR (qPCR) or digital droplet PCR (ddPCR) approaches. Error-corrected or barcoded sequencing, using molecular identifiers, can increase the sensitivity of NGS to 10^−5^ for MRD purposes [98]. In addition to these molecular MRD approaches, every piece of information should still be correlated with flow cytometry, cytomorphology/histopathology, and cytogenetic findings. For the MDS cases, molecular evaluation allows a more accurate prognostication, as compared to cytogenetics alone [37].

The advances in the knowledge of the clinical significance of the genetics and molecular alterations in MDS lead to newer risk stratification models. The Personalized Prediction Model for stratifying patients with MDS is a dynamic stratification tool which combines clinical and genomic data in predicting the risk of death and the leukemia transformation of MDS patients, which can upstage or downstage the disease. The addition of molecular data in stratifying MDS patients allows for a more accurate prediction of disease behavior and treatment-directed strategies. A 24 gene panel, in addition to clinical data, was used to analyze the MDS cases using machine learning techniques. A significant impact in OS and in AML transformation was seen, when the molecular data were taken heavily into consideration in the stratification scheme, including not only the variant involved but also the number of mutations identified [95].

The proposal of a newer Molecular International Prognostic Scoring System (IPSS-M), using a combination of clinical, cytogenetic, and genetic parameters, in which the primary endpoint is leukemia-free survival, was presented. The IPSS-M model is a refinement of the IPSS-R, in which the variables consist of hemoglobin levels, platelet count, bone marrow blast count, IPSS-R cytogenetic category, and the presence of mutations in 31 genes (Bernard et al.; ASH Abstract 2021). Using this model and cut-offs, a six-risk category was redefined: very low; low; moderate low; moderate high; high; and very high (Figure 6).

## 6. Machine Learning and Future Directions

Artificial intelligence (AI) is a field that focusses on automating intellectual tasks normally performed by humas, while machine learning (ML) and deep learning (DL) are the specific methods for achieving this goal. ML is a subfield of AI, which allows pattern recognition in high-dimensional space. DL, on the other hand, is a subset of ML, in which artificial neural networks (ANNs) are utilized in learning complex functions [100]. The convolutional neural network (CNN) is a form of ANN that preserves the spatial relationship between pixels in an image [101,102].

Due to several unavoidable issues related to interobserver variability in the morphologic evaluation of MDS, ML strategies can provide a supplementary tool to minimize the degree of inherent subjectivity. The identification of BM dysplasia to diagnose MDS can sometimes be challenging, and therefore the ML approach may aid in establishing the diagnosis. AKIRA is the first CNN-based AI system that was developed to identify the decreased granules in neutrophils, which is one of the most common forms of dysplastic change seen in MDS (Figure 7). The system reports a markedly high predictive accuracy of 97.2% [103].

As a corollary to advances in molecular genetics and the morphological assessment of dysplasia, quantitative image analysis improves the accuracy and reproducibility of MDS diagnosis. One approach is the use of real-time deformability cytometry (RT-DC) imaging, to standardize and measure the morphological and mechanical properties of single cells in the blood or bone marrow. It utilizes automated image analysis and ML to characterize thousands of cells. The images are captured by a high-speed camera and each of the cell images are analyzed in real-time to obtain the cell area, length and height, aspect ratio, deformation, inertia ratio, and porosity, and therefore aid in cellular designation [104]. Another platform that can be used for the ML approach in the diagnosis of MDS is imaging flow cytometry (IFC). IFC integrates the technology behind the conventional multiparametric flow cytometry with the DL approach of visual assessment by microscopy. IFC contains a detection system that simultaneously generates up to 12 digital images, which are a combination of bright field and fluorescent microscopy images. The collected data will be analyzed by a computer algorithm which can quantitate the size and shape of the cells and cellular compartments [105].

## 7. Conclusions

MDS is a clonal myeloid neoplasm which can be difficult to diagnose by conventional methods. Screening clinical and morphologic evaluation, as well as flow cytometric findings, and the detection of genetic lesions by newer technical approaches and modalities, in particular, cytogenetics/FISH and mutational testing by NGS platforms, provide a comprehensive step-wise approach for the diagnosis, prognostication and therapeutic decisions. All in all, the hematologists, oncologists and the hematopathologists should remain in close coordination and in constant interaction for holistic patient care. Such combined efforts can lead to an optimized use of the newer diagnostic modalities for the benefit of patients with myeloid malignancies and their overall welfare.

## Figures and Tables

**Figure 1 diagnostics-12-01581-f001:**
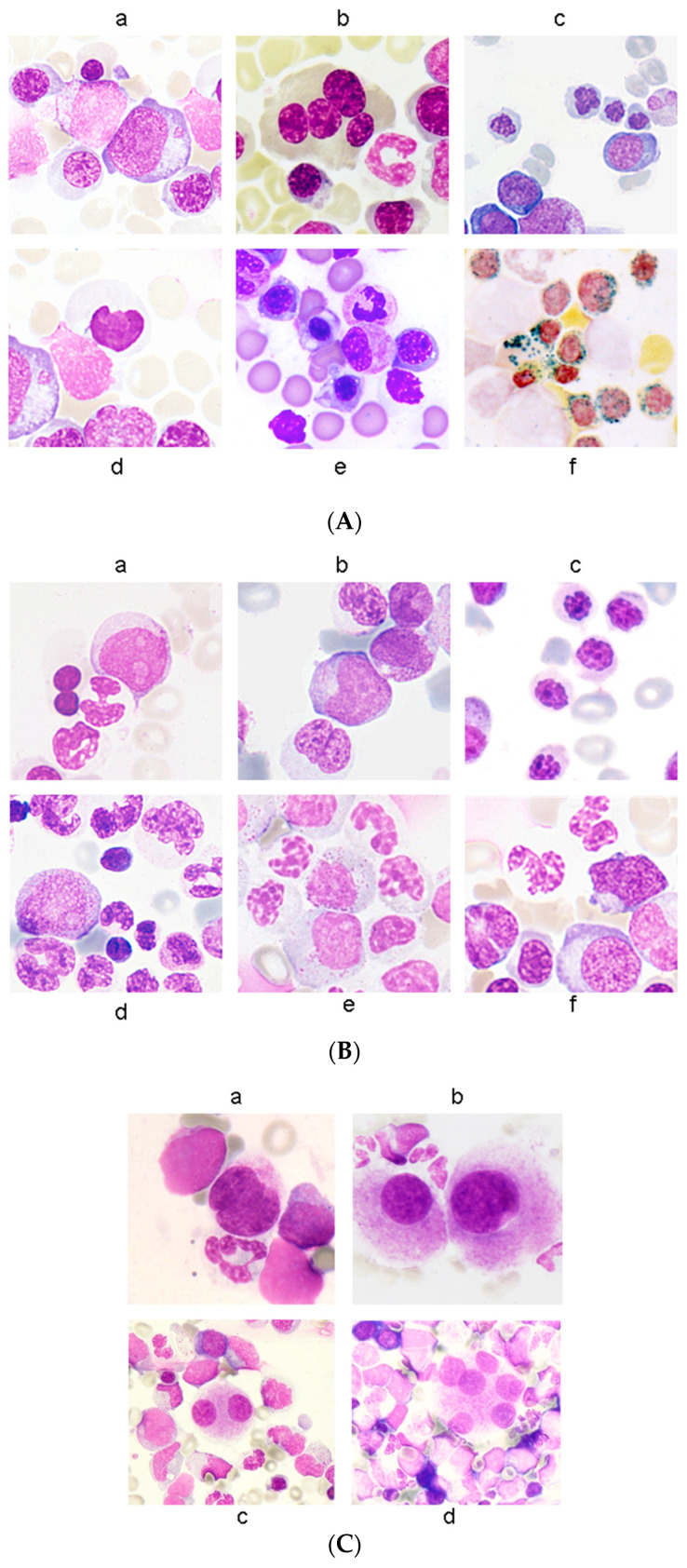
(**A**) Erythroid dysplasia (100×). (**a**) megaloblastosis; (**b**) multinuclearity; (**c**) nuclear lobulation; (**d**) pyknosis; (**e**) defective hemoglobinization and cytoplasmic fraying; (**f**) ring sideroblasts. Adapted from Della Porta et al. [9]; (**B**) Granulocytic dysplasia (100×). (**a**) myeloblast; (**b**) Auer rod; (**c**) hypolobulation; (**d**,**e**) abnormal nuclear shape; (**f**) hypogranulation. Adapted from Della Porta et al. [9]; (**C**) Megakaryocytic dysplasia (100×). (**a**) micromegakaryocyte; (**b**) monolobated megakaryocyte; (**c**) small binucleated megakaryocyte; (**d**) megakaryocyte with multiple separated nuclei. Adapted from Della Porta et al. [9].

**Figure 2 diagnostics-12-01581-f002:**
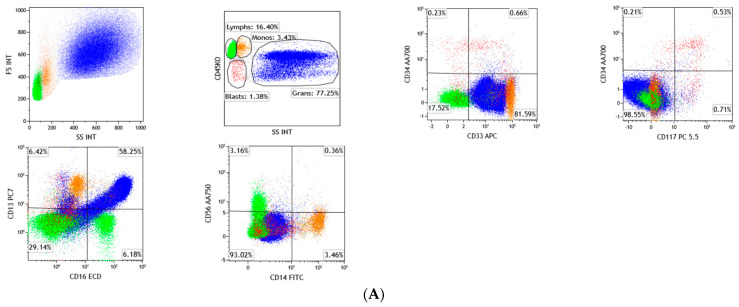
(**A**) Shows normal maturational pattern of nucleated hematopoietic cellular populations. Regenerating myeloblasts showing intact CD34 and CD117 expression and heterogeneous CD33 positivity. Maturing granulocytic precursors showing the classic “Nike swoosh” pattern in the CD13/CD16 plot, and monocytes showing intact expression of CD14, without aberrant antigen expression. *Lymphocytes in green, monocytes in orange, granulocytes in blue, blasts in red*; (**B**) Case of MDS showing myeloblasts with loss of CD117 and with concurrently bright CD33 intensity. Prominent abnormal granulocytic maturational pattern in CD13/CD16 plot, losing the classic “Nike swoosh” pattern. In this MDS case, the granulocytes and monocytes also show aberrant CD56 expression; *Lymphocytes in green, monocytes in orange, granulocytes in blue, blasts in red*.

**Figure 3 diagnostics-12-01581-f003:**
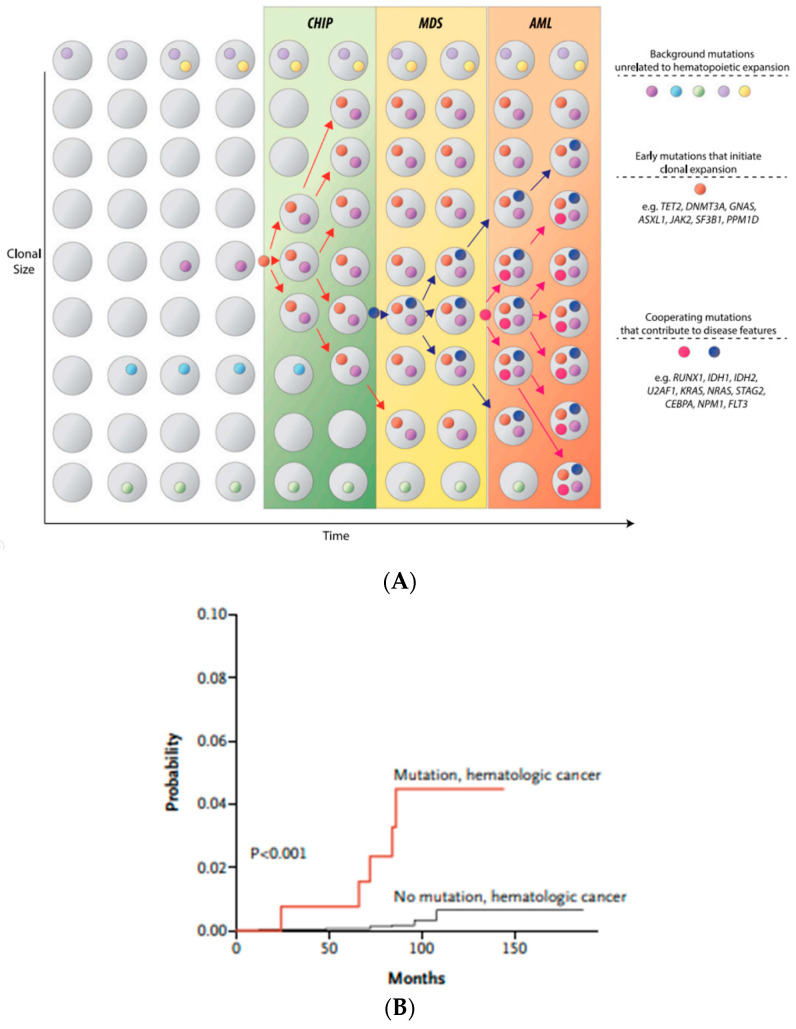
(**A**) CHIP as a precursor to hematologic neoplasms. Sequential acquisition of somatic mutations leading to clonal instability and survival advantage of mutated cells leading to expansion of neoplastic clones. Adapted from Steensma et al. [85]; (**B**) Cumulative incidence of hematologic malignancies. With increasing number of somatic mutations acquired during aging (CHIP), there is a proportional increase in the risk of developing hematologic malignancies. Adapted from Jaiswal et al. [80].

**Figure 4 diagnostics-12-01581-f004:**
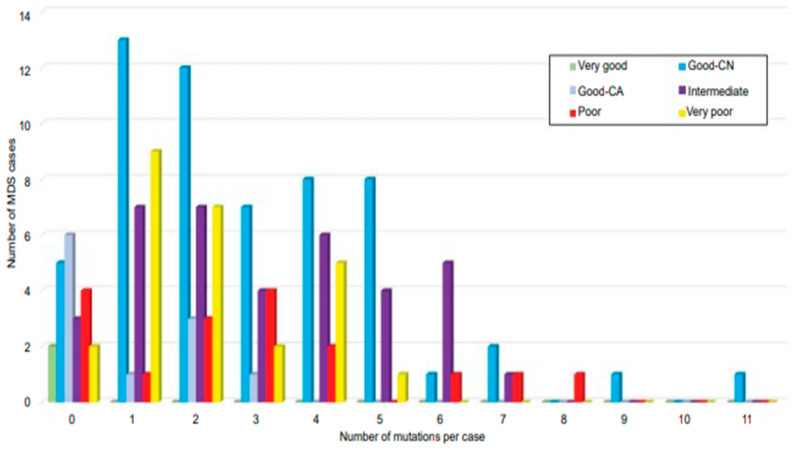
Correlation of number of co-mutations in cytogenetic prognostic-stratified MDS. Two or more mutations identified in 69% of cytogenetically normal MDS (blue), occurring in up to 11 mutant genes in a single patient. Adapted from Tria et al. [58].

**Figure 5 diagnostics-12-01581-f005:**
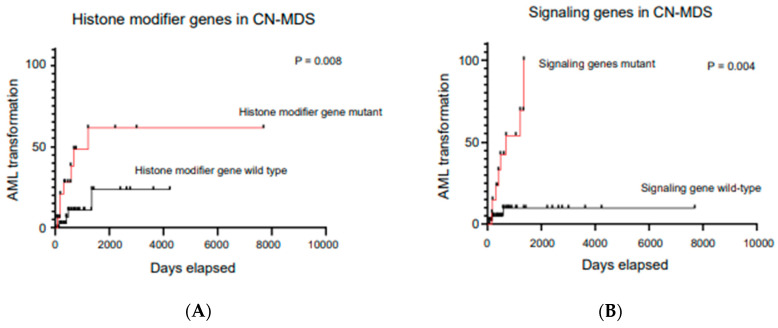
(**A**,**B**). Data showing AML transformation occurring significantly faster in patients who acquire histone modifier gene and signaling gene mutations at initial diagnosis. Adapted from Tria et al. [58].

**Figure 6 diagnostics-12-01581-f006:**
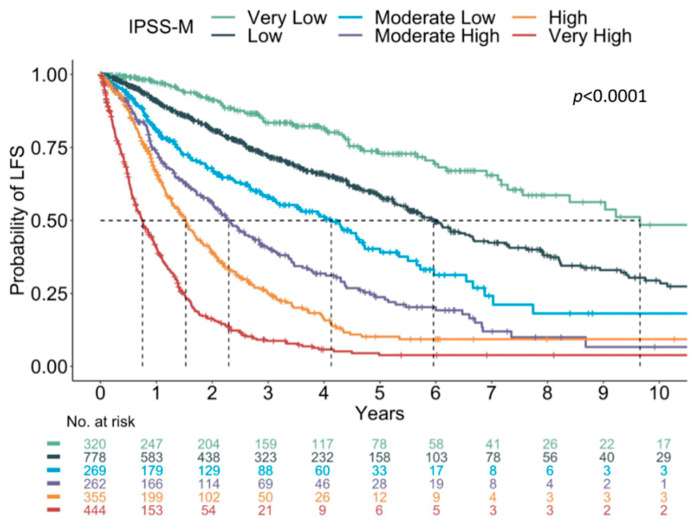
Kaplan–Meier curves in IPSS-M showing significant differences in probability of leukemia-free survival in each subgroup. Adapted from Bernard et al. [99].

**Figure 7 diagnostics-12-01581-f007:**
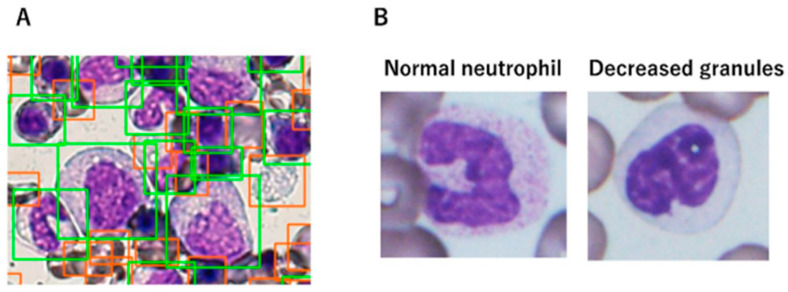
Detection of dysplasia by AI. (100×) (**A**) The detectors distinguish the cells of interest (green boxes include all nucleated cells and red boxes include red blood cells, platelets, and debris); (**B**) Normal mature neutrophil with adequate cytoplasmic granules (**left**) and granulocytic hypogranularity in a mature neutrophil (**right**). Adapted from Mori et al. [103].

**Table 1 diagnostics-12-01581-t001:** The Comprehensive Cytogenetic Scoring System (CCSS) for myelodysplastic syndromes. Adapted from Schanz et al. [22].

Prognostic Subgroup	Defining Cytogenetic Abnormalities
Very good	Loss of Y chromosome
	del(11q)
Good	Normal
	del(5q)
	del(12p)
	del(20q)
	Double, including del(5q)
Intermediate	del(7q)
	Gain of chromosome 8
	Gain of chromosome 19
	Isochromosome 17q
	Single or double abnormalities not specified in other subgroups
	Two or more independent non-complex clones
Poor	Loss of chromosome 7
	Inv(3), t(3q) or del(3q)
	Double including loss of chromosome 7 or del(7q)
	Complex (three abnormalities)
Very poor	Complex (>three abnormalities

**Table 2 diagnostics-12-01581-t002:** The Revised International Prognostic Scoring System (IPSS-R) score values for myelodysplastic syndromes. Adapted from Greenberg et al. [5].

Prognostic Variable	Score Values						
	0	0.5	1	1.5	2	3	4
Karyotype (CCSS)	Very good	-	Good	-	Intermediate	Poor	Very poor
BM blast percentage	≤2%	-	>2% to <5%	-	5–10%	>10%	-
Hemoglobin concentration (g/dL)	≥10	-	8 to <10	<8	-	-	-
Platelets (×10^9^/L)	≥100	50 to <100	<50	-	-	-	-
Absolute neutrophil count (×10^9^/L)	≥0.8	<0.8	-	-	-	-	-
Five risk groups are defined on the basis of total score of the parameters listed above: Very low: ≤1.5 Low: >1.5 to 3 Intermediate: >3 to 4.5 High: >4.5 to 6 Very high: >6							
- Indicates not applicable							

**Table 3 diagnostics-12-01581-t003:** Recurrent chromosomal abnormalities and frequencies in MDS. Adapted from Swerdlow et al. [6].

Chromosomal Abnormality		Frequency	
		MDS Overall	Therapy-Related MDS
Unbalanced	Gain of chromosome 8 *	10%	
	Loss of chromosome 7 or del(7q)	10%	50%
	del(5q)	10%	40%
	del(20q) *	5% to 8%	
	Loss of Y chromosome *	5%	
	Isochromosome 17q or t(17p)	3% to 5%	25% to 30%
	Loss of chromosome 13 or del(13q)	3%	
	del(11q)	3%	
	del(12p) or t(12p)	3%	
	idic(X)(q13)	1% to 2%	
Balanced	t(11;16)(q23.3;p13.3)		3%
	t(3:21)(q23.2;q22.1)		2%
	t(1:3)(p36.3;q21.2)	1%	
	t(2;11)(p21;q23.3)	1%	
	inv(3)(q21.3;q26.2)/t(3;3)(q21.3;q26.2)	1%	
	t(6;9)(p23;q34.1)	1%	

* As a sole cytogenetic abnormality in the absence of morphological criteria, gain of chromosome 8, del(20q), and loss of Y chromosome are not considered definitive evidence of MDS, in the setting of persistent cytopenia of undetermined origin, the other abnormalities are considered presumptive evidence of MDS, even in the absence of definitive morphological features.

## Data Availability

Data sharing not applicable.

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
