# Peer review of "Myelodysplastic Syndrome: Diagnosis and Screening"

_diagnostics, 2022, doi:10.3390/diagnostics12071581_

Round 1
Reviewer 1 Report
Comprehensive review on the state of art in MDS diagnosis.
I have no major concerns for the pubblication of the review. Some spelling errors should be fixed. e.g. line 320 (Isochromosome 17q instead of Isochromosome 7q) or line 242 (its instead of it's).
Author Response
June 7, 2022
Dear Martyna,
We are resending the revised version of our manuscript entitled “Myelodysplastic Syndrome: Diagnosis and Screening”. We adequately addressed all the reviewers’ comments and suggestions. We hope you find our edited version more appealing to the readers.
• Spelling errors are fixed.
• Additional discussions are included regarding the utility of various gene mutations and their predictive values in MDS. Also added are current literature on machine learning strategies, recent personalized prognostic prediction of MDS and proposal regarding IPSS-M. More discussions on CHIP and CCUS are also included.
Thank you and we are looking forward to your positive response.
Guang Fan, MD, PhD
Medical Directory, Department of Hematopathology
Oregon Health & Science University
Portland, Oregon, USA
Francisco Tria IV, MD
Hematopathologist
St. Luke’s Medical Center
Taguig, Metro Manila, Philippines
Daphne Ang, MD
Hematopathologist
St. Luke’s Medical Center
Taguig, Metro Manila, Philippines

Reviewer 2 Report
This is a comprehensive review on the diagnosis and screening for MDS.
Specific issues that has to be addressed:
1. There is paucity of discussion on the utility of various gene mutations and their predictive value for the diagnosis of MDS.
2. Please also include current literature on the use of machine-learning strategies in the diagnosis and screening of MDS.
3. Please include and discuss the personalized prognostic prediction of MDS (JCO 2021) and the recent abstract on IPSS-M (ASH 2021)
4. Please expand on the discussion on CHIP and CCUS.
Author Response

(The authors gave the same response as above.)

Round 2
Reviewer 2 Report
All issues have been addressed.
Author Response
We appreciate your comments and efforts to review our manuscript. Attached is our final version for publication.
